# Peer review of "Evolution of Microstructure at the Surface of 40CrNiMo7 Steel Treated by High-Current Pulsed Electron Beam"

_coatings, doi:10.3390/coatings10040311_

Round 1

Reviewer 1 Report

Here is the list of my comments:

- I consider that for the reading of the article, it is advisable to use tables when entering the data obtained, such as for the analysis of chemical composition. I consider that so much text often does not help reading process.

- For the chemical composition the error of the measurements should be included.

- In the Figure 1 the scale bar is not well read.

- Also, some spacing should be checked both after the dot, and to separate paragraphs.

- Finally, I recommend improving the conclusions.

Reviewer 2 Report

Comments below.

Line 70 - is this an observation or a known fact ? It should be referenced properly

Figure quality needs work. Please fix the figures.

What causes the surface roughness increase after pulsing ? Can you provide a more physics based explanation ?

Also there are other surface roughness methods. Can you discuss some other methods for measuring the roughness ?

Fig4 bottom parts are NOT clear at all

TEM analysis needs more detail. 1 pictures does not provide \enough material to make a solid claim. Have the authors looked at multiple areas and they all show the same pattern ? 

Please discuss the XRD patterns. The identified peaks seem to be just random noise in XRD rather than actual results.

Combine (b) and (c) for Fig 8/. Why did the corossion performance change after 50 pulses ? Is it possible to define a relationship (analytical) and suggest a best value based on energy or pulse quantities for optimal corossion resistance ?

Reviewer 3 Report

The paper is very clear and well written. I have some comments to do, that are regarding to minor changes:

  • There are some typos, such as in line 59: 26v instead of 26 V.
  • Line 26. References are not explained in detail, therefore, their need could be questioned. The authors are encouraged to develop the introduction further, mentioning specific comments about the references included in it.
  • Also about the introduction, mechanical methods should also be mentioned. There have been very interesting results with ultrasonic-vibration-assisted ball burnishing, such as in this one:

https://doi.org/10.1016/j.jmatprotec.2018.12.022

Authors are intensively encouraged to include it in the introduction, so that authors can offer a complete review of the state of the art.

  • Line 87: Ra is called average surface roughness
  • Line 90: Authors say "The surface roughness was 0.48 cm and 0.36 cm after 3 and 8 pulses...", however, this is not coherent with the results (and makes no sense in terms of absolute magnitude). Could it be clarified?
  • Line 90: Then it is stated: "After 15 pulses treatment, the modified surface became smoother with the roughness 92 less than 0.2m". Again, there is a conceptual mistake here to be clarified
